# Induced Pluripotent Stem Cells in Birds: Opportunities and Challenges for Science and Agriculture

**DOI:** 10.3390/vetsci11120666

**Published:** 2024-12-19

**Authors:** Nousheen Zahoor, Areej Arif, Muhammad Shuaib, Kai Jin, Bichun Li, Zeyu Li, Xiaomeng Pei, Xilin Zhu, Qisheng Zuo, Yingjie Niu, Jiuzhou Song, Guohong Chen

**Affiliations:** 1Joint International Research Laboratory of Agriculture and Agri-Product Safety, Ministry of Education of China, Yangzhou University, Yangzhou 225009, China; nousheenzahoor99@gmail.com (N.Z.); yubcli@yzu.edu.cn (B.L.); mx120230901@stu.yzu.edu.cn (Z.L.); 008617@yzu.edu.cn (X.P.); mz120231580@stu.yzu.edu.cn (X.Z.); 006664@yzu.edu.cn (Q.Z.); niuyi@yzu.edu.cn (Y.N.); ghchen@yzu.edu.cn (G.C.); 2Key Laboratory of Animal Breeding Reproduction and Molecular Design for Jiangsu Province, College of Animal Science and Technology, Yangzhou University, Yangzhou 225009, China; shoaibwzr@gmail.com; 3Institutes of Agricultural Science and Technology Development, Yangzhou University, Yangzhou 225009, China; 4College of Animal Science and Technology, Yangzhou University, Yangzhou 225009, China; areej.arif.2024@gmail.com; 5College of Bioscience and Biotechnology, Yangzhou University, Yangzhou 225009, China; 6Department of Animal & Avian Sciences, University of Maryland, College Park, MD 20742, USA; songj88@umd.edu

**Keywords:** induced pluripotent stem cells, reprogramming factors, signaling pathways, genetic considerations, applications

## Abstract

Induced pluripotent stem cells have revolutionized biological research. These are adult cells that can be reprogrammed to differentiate into any cell type in the body. Although iPSCs have been widely studied in mammals, recent progress has extended their use to avian species, particularly chickens. Chicken iPSCs provide a unique model for investigating avian development, pathology, and genetics. Through the manipulation of these cells, researchers may get insights into cellular differentiation, disease progression, and possible tissue healing mechanisms. The development of efficient reprogramming methods, both viral and non-viral techniques, has significantly improved the generation of chicken iPSCs. In addition, a better understanding of the underlying molecular mechanisms, such as key signaling pathways and transcription factors, has driven the science forward. There are many uses of chicken-induced pluripotent stem cells. They may be used in the modeling of human diseases, the formulation of new drugs, and the engineering of genetically modified birds for agricultural purposes. As research proceeds, we are expecting more revolutionary discoveries and new uses of chicken iPSCs.

## 1. Introduction

The only cells in an organism that could do any other sort of cell until 2006 (except sperm or egg) were known as embryonic stem cells, ESC. ESCs can divide indefinitely and form chimeras (to ensure long-term transmission of genetic material to offspring) by [1] bypassing spermatocytes. So far, such long-lived embryonic stem cells have only been proven to exist in rats and mice [2]. When it comes to cultured embryonic stem cells (ESCs) from species other than humans, mice, and rats, the use of specified culture conditions poses the initial challenge. In 2006, a new method was created to produce stem cells that resemble embryonic cells [1]. This method involves transforming specialized adult cells into iPSCs [1]. iPSCs have been created in the laboratory by reprogramming already differentiated cells. iPSCs offer significant potential in cellular therapeutic reprogramming as well as in early development techniques, as they can develop into a variety of corresponding cellular lineages, containing the three germ layers. Since the advancement in iPSC generation in 2006 [1]. There are several benefits of using iPSCs as opposed to other stem cell types such as ESCs and MSCs [3]. Numerous investigations have shown that cellular pluripotency may be induced by transcription factor overexpression. Induced pluripotent stem cells can self-renew and differentiate into any kind of adult cell, much as ESCs, as shown in Figure 1. In applications as models for wound healing and regenerative medicine, iPSCs offer distinct advantages over other stem cell varieties. The moral dilemmas raised by using embryonic stem cells do not apply to iPSCs, because adult somatic cells rather than embryos are used to obtain them [4].

It has long been known that using avian embryonic models, important new discoveries on developmental biology, such as organ function [5,6], the course of diseases (such Pompe syndrome) [7], eye abnormalities [8], and many other topics [9,10], can be made. Bird species have the advantage of being quite small and having easy access to the embryo for modification. It is possible to transplant cells and tissues, even entire spinal column segments, into the avian embryo and track its progress in real time [11]. Furthermore, because cells in the quail-chicken chimera can be easily tracked, it is a desirable and popular model for investigations on cell fate and developmental patterning [12]. Chickens, also known as Gallus gallus, are valuable sources of protein because they are inexpensive and have few limitations related to faith. In addition, hens are frequently employed as model organisms in scientific research. Chickens have also been used as a model organism to study the evolution in amniotes, which includes humans [13]. Therefore, the genetic transformation of chickens is very important in the advancement of scientific research and industrial development.

The reasons why chicken is a good model system to study iPSCs include:One such advantage of using the chicken embryo as a model organism is that it allows visualization of and measurement tracking in life growth.Their embryos are easily accessible and quite big, both of which aid in the ability to manipulate and watch developmental activation—reasons they have been frequently used for decades as a “model organism” by biologists studying embryonic development [14].Stern (2005) found that the chicken embryo, a traditional model system for morphogenesis and organogenesis research, has yielded important insights into vertebrate development [15].Also, chickens are among the most completely sequenced vertebrate genomes and represent a pharmacogenomic model species that is particularly conducive to effective transgenesis or CRISPR/Cas9 gene editing protocols, which could assist in both forward genetic screening for functional annotation of genes involved in instructing lineage-specific mammalian cellular functions [16].Chickens (to study pathways of cellular reprogramming in iPSC research).Chickens are used as a reprogramming paradigm by Rossello and Torres-Padilla (2011), who have also demonstrated the importance of species-specific factors that could contribute to differences in yet uncharacterized cellular responses [17].Liu et al. (2017) generated chicken iPSCs from somatic cells that could differentiate into all three germ layers, such as human and non-human primate (NHP)-derived iPSCs [18].Li et al. showed that chicken iPSCs are utilized for regenerative purposes and highlighted valuable aspects of these cells as tools to decipher tissue development [19].

Here, we aim to provide a comprehensive review of the recent advances in chicken iPSCs research. It will discuss how they each work in combination with the other, as well as their applications to developmental biology, disease modeling, and possibly regenerative medicine. It will offer insights for future research directions and treatment advancements.

**Figure 1 vetsci-11-00666-f001:**
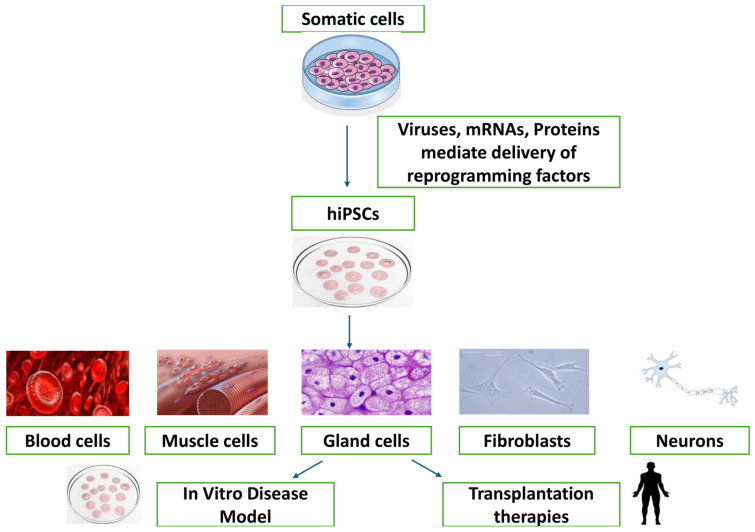
Process of Reprogramming iPSCs [19] reprinted and modified with permission from [19]. Copyright 2016, Elsevier.

## 2. Historical Development of iPSCs in Chickens

The conception of iPSC technology in 2006 made it possible to establish PSCs using an alternative protocol that can be applied to not only domestic animals but also non-native species sophistically [20]. The first stream was nuclear transfer reprogramming. John Gurdon (1962) announced the creation of tadpoles from unfertilized eggs using intestinal cells from adult frogs for the nucleus. More than thirty years later, Ian Wilmut and his colleagues made a significant announcement about the production of Dolly, who was the first animal produced by replicating mammary epithelial cells by somatic cloning. Successful somatic cloning experiments have shown that it is possible to reprogram the genetic material in somatic cell nuclei within oocytes. Furthermore, these investigations have revealed that even specialized cells possess all the required genetic material to generate fully formed animals. Takashi Tada’s team (2001) provided evidence that embryonic stem cells (ESCs) contain components capable of reprogramming somatic cells. In 1987, researchers made the groundbreaking discovery that when the Drosophila transcription factor Antennapedia is generated in abnormal locations, it leads to the development of legs rather than antennae. That year, researchers discovered that fibroblasts converted into myocytes by using mammalian transcription factors, MyoD. These findings opened the door for the idea of a transcription factor known as a “master regulator”, which controls and dictates the development of certain lineages Figure 2.

Since the initial derivation of mouse ESCs in 1981, culture conditions developed by Austin Smith and others have enabled the long-term maintenance of pluripotency. Mouse ESCs need LIF to be viable. Ever since human embryonic stem cells were initially generated, culture conditions that are favorable for basic fibroblast growth factor (bFGF) have been discovered. The first iPSCs were generated through retroviral transduction of four genes—Sox2, Oct4, c-Myc, and Klf4—into the genome of a donor cell [21]. The work of Pain et al. (1996) represented a milestone in this field, obtaining chicken embryonic stem cells (ESCs) and paving the way for further progress [22]. This was a groundbreaking contribution and served as the basis for understanding pluripotency, which is like that in mammals or long-term culture of stem cells in avian species. These initial studies demonstrated the ability of chicken cells to be pluripotent, representing a key advance in iPSC technology. One of the milestones in this line has been the generation, for the first time ever, of chicken iPSCs from embryonic fibroblasts via reprogramming with defined factors [23]. Liu et al. However, Dai et al. (2017) showed that chicken embryonic fibroblasts can be reprogrammed into iPSCs by the combination of Yamanaka factors OCT4/SOX2/KLF4/c-MYC [16]. It was the first report of avian iPSCs and significantly advanced avian genetic research and biotechnology. Rossello and Torres-Padilla evolved further by analyzing the efficiency of reprogramming along with its underlying mechanisms in avian species [24]. Their work shed more light on the molecular pathways responsible for cellular reprogramming, revealing both well-conserved and species-specific properties between avian and mammalian systems. These emphasized how important it is to understand species-specific components that regulate iPSC reprogramming fidelity. Thanks to these early studies into this phenomenon and the discoveries made because of it, chickens have been an excellent model for iPSC research, leading us to discover exciting new applications in biotechnology and medicine. These studies have shown key features of pluripotency.

**Figure 2 vetsci-11-00666-f002:**
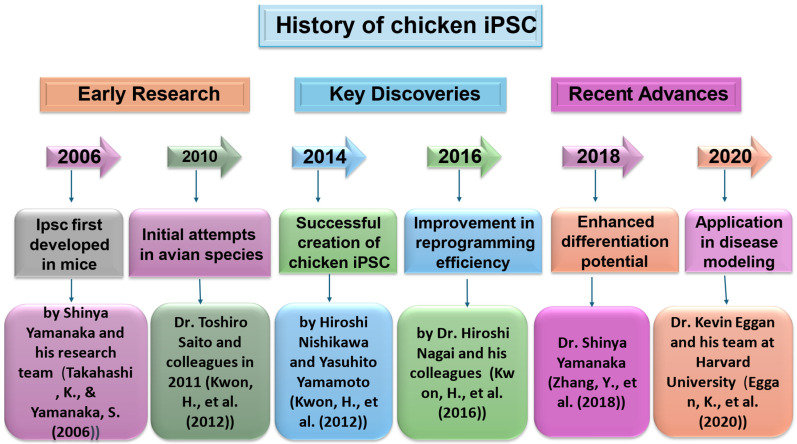
Historical background of iPSC production [1,25,26,27].

## 3. Techniques for iPSC Induction in Chickens

Scientists have tried various ways to reprogram chicken iPSCs, each with its pros and cons [28]. One common method uses viral vectors such as retroviruses and lentiviruses to reprogram chicken body cells with transcription factors such as Oct4, Sox2, Klf4, and c-My [29]. This approach works well for reprogramming, but it might cause insertional mutagenesis as virus vectors enter the host’s DNA [30]. Non-viral techniques such as the PiggyBac transposon system and episomal plasmids can lower the risk of genetic integration as shown in Figure 3 [31]. Gene delivery methods that are integrated are more effective than those that are not, but they are less safe due to the chance of insertional mutagenesis [32].

Recent studies show that viral integrative techniques have an impact on the creation of chicken iPSCs [39]. These techniques involve adding key reprogramming factors such as Oct4, Sox2, Klf4, and c-Myc (OSKM) to somatic cells using retroviral and lentiviral vectors [40]. The first iPSC research used retroviral vectors to insert recalculating variables into the host cells’ genes. During reprogramming, retroviral transgenes often become inactive, while DNA and histone methyltransferases start working. The reprogramming process is often incomplete, leading to iPSC lines that need outside help and cannot turn on important endogenous genes. Additionally, somatic cells derived from iPSCs may be affected by viral transgenes or their rejuvenation, which may interfere with a cell’s proper growth and potentially cause the formation of cancerous growths [31]. Using retroviruses to boost the activity of transcription factors is an effective and simple technique. Retroviruses specifically aim to actively divide somatic cells so that their genetic material may be effectively incorporated into the genome of the host cell [41].

Lentiviruses are RNA viruses belonging to the Retroviridae family that have a single-stranded structure. These viruses are very efficient in transferring genes because they may infect cells that are actively dividing as well as those that are not. The lentiviral genome integrates into the host cell’s genome during reverse transcription. Lentiviruses are often used when stable expression in target cells is needed since they can integrate into the host genome and allow for long-term expression in vitro. This characteristic is absent from episomal vectors [42]. Although lentiviruses have a high transduction efficiency, multiple proviral integrations and insertional mutagenesis are possible. These may lead to misplaced transcripts and abnormal alternative splicing [40]. By controlling lentiviral vector expression, doxycycline may reduce the likelihood of continuous transgenic expression and make it easier to choose fully reprogrammed iPSCs. This is because the reprogramming cells depend on external stimuli for expression, and they stop proliferating when doxycycline is withdrawn [43].

In 2008, Stadtfeld et al. successfully produced the initial integration-free iPSCs using nonintegrative adenoviruses. These iPSCs were derived from fully developed mouse hepatocytes [31]. In 2009, Zhou and Freed successfully produced iPSCs from human fibroblasts without including transgenes. They used adenoviral vectors to do this. The process of producing adenoviruses takes a long time, and the reprogramming process is not as effective as it can be when using lentiviruses and retroviruses. These are just a few of the disadvantages of using adenoviral vectors to produce iPSCs. The non-integrative Sendai virus (SeV), possessing a negative single-stranded RNA, is an additional effective viral vector for introducing genes to diverse somatic cells. However, it is challenging to remove SeV vectors from cells because of their constant replication. Furthermore, the addition of transgenic protein has a high degree of susceptibility to the viral vector’s RNA copy [41]. Adenoviruses are a type of virus that do not have an outer envelope and contain double-stranded genomic DNA. They can cause the temporary expression of a transgene [44]. These viruses are non-integrative and are useful for producing iPSCs by acting as expression vectors. The efficiency of reprogramming with adenoviral vectors is significantly lower compared to lentiviruses or retroviruses, with rates of 0.001%–0.0001% in mice and 0.0002% in human cells [42]. Although adenoviruses are not very good at carrying genetic material, they may be used to transport large gene inserts by using gutless adenoviruses (GLAd). Nevertheless, GLAd viruses are troublesome since they need the presence of another virus to co-infect, which complicates subsequent cleaning procedures. The presence of coxsackie and adenovirus receptors on the target cells is necessary for the adenoviruses to transfer genes more successfully [36].

The Paramyxoviridae family of RNA viruses includes the non-pathogenic Sendai virus. It is a single-stranded, encapsulated RNA with a negative sense. SeV does not integrate into the host genome because its life cycle does not include a DNA intermediary. It is hence ideal for creating iPSCs devoid of transgenes. The vectors of the Sendai virus (SeVVs) replicate as ribonucleoprotein (RNP) complexes, and without going through the DNA phase, transcription takes place in the cytoplasm of the host cell. Full iPSC reprogramming is reportedly made achievable by the persistent production of reprogramming factors made possible by the RNP-based replication of SeVs in host cells [45]. In 2009, the Hasegawa Group published the first study on how well-executed reprogramming factors based on SeV can be used to reprogram human fibroblasts. Instead of combining the four typical Yamanaka factors—Oct4, Sox2, Klf4, and c-Myc—into one virus, the scientists developed unique SeV structures for every component of reprogramming. This approach was chosen because it offers more control over the reprogramming factor’s stoichiometry, reduces the danger of carcinogenesis, and has fewer side effects than in a single virus; all four reprogramming factors are expressed simultaneously [44].

Non-integrating, non-viral systems utilize episomal vectors or plasmids to temporarily express reprogramming agents. The reprogramming factors’ cDNA, such as Oct3/4, Sox2, Klf4, and c-Myc, are included in these vectors or plasmids [31]. The use of episomal vectors may currently be the most appropriate reprogramming technique for iPSC usage in medicine, as iPSCs created using this technique do not show indications of their genomes by plasmid insertion.

Apart from chromosomal DNA, episomes are extra DNA molecules that can multiply on their own within the cell. Reprogramming factors may be temporarily delivered into somatic cells via episomal vectors, such as plasmids. Episomal vectors, unlike retroviruses and lentiviruses, offer greater ease of use and ensure dependable gene expression without the need for genomic inclusion. Episomal vectors exhibit temporary expression and so require multiple transfections, resulting in reduced effectiveness of reprogramming using this approach [46]. Because episomal vectors are not integrative, cells need repeated passaging to eliminate or dilute them and subsequent cell divisions, which reduces the possibility of insertional mutagenesis and persistent production of pluripotency factors. However, given the DNA makeup of these vectors, it is impossible to fully rule out the possibility of genomic integration [47].

The PiggyBac transposon system is a highly efficient and effective technique for converting chicken somatic cells into iPSCs [48]. This approach is particularly advantageous because it can neatly remove the transposon. The PiggyBac method employs a transposase enzyme that facilitates the insertion and removal of DNA sequences bordered by inverted terminal repeats (ITRs) [49]. The PiggyBac transposon technology was employed to introduce reprogramming factors (Oct4, Sox2, Klf4, and c-Myc) into chicken fibroblasts [50]. The transposons containing these components were effectively inserted into the genetic material of the host, resulting in the successful alteration of the cells’ programming [51]. The integration process is mediated by the transposase enzyme. The enzyme recognizes the ITRs and integrates the transposon into TTAA spots on chromosomes [52]. These spots neutrally occur in the genome and therefore reduce the possibility of insertional mutagenesis [53]. Another big advantage of the PiggyBac system is that genetic material is removed without leaving any footprints. This would mean that after the reprogramming and setup of iPSCs, we could bring back the transposase to cut out the transposon. This would basically leave the genome free from any remaining foreign sequences. This feature specifically targets a significant safety issue related to genomic changes, which is the possibility of insertional mutagenesis and interference with the normal functioning of genes within an organism [54].

In 1987, Escherichia coli was found to have repetitions of CRISPR. Their formal term was changed to Clustered Regularly Interspaced Short Palindromic Repeats (CRISPR) in 2002 [55]. It was not until 2010 that its function in DNA splicing and the bacterial defense system was discovered. The molecular makeup of the CRISPR system was identified in 2012 by Jennifer Doudna and Emmanuelle Charpentier. They were awarded the 2020 Nobel Prize in Chemistry for this [56]. Thereafter, several groups modified the construct of the CRISPR/Cas9 system and applied it to human cells [57]. ZFNs and TALENs have not been used much in gene editing compared to the CRISPR system due to their improved specificity, efficiency, and ease of reprogramming. The CRISPR system now has twenty-one subtypes and six types. These include the widely used CRISPR/Cas9 system, applied in RNA imaging and rapid nucleic acid detection, and the newly discovered CRISPR/Cas12a and CRISPR/Cas13 systems [58]. Based on the composition of the effector of the prokaryotic immune system, the CRISPR system could be divided into two categories. The Multiple effector proteins, includes types I, III, and IV, and are one of the several categories under which the CRISPR/Cas system is divided. The second category is the single multi-domain protein effectors, which are the most employed category so far. Types II, V, and VI belong to this category. The CRISPR/Cas9 system largely consists of the Cas9 protein. The other components used in this system include the CRISPR-derived RNA, or crRNA, and the tracrRNA, which is the trans-activating crRNA that help in precise targeting. Complementary tracrRNA pairs with crRNA to form a double-stranded RNA molecule, which has a parallel structure. Subsequently, the complex is joined by the Cas9 protein to perform targeting of specific sections of DNA. The Cas9 protein recognizes sequences called protospacer adjacent motifs, and it cuts the DNA in specific places using endonuclease activity, hence creating DNA double-strand breaks. After that, a DNA strand break triggers cells to initiate the process of repairing the broken strand. In the repair process, DNA may be inserted, deleted, or replaced, resulting in a modification of the DNA target sequences. In modern applications of CRISPR tools to edit DNA, crRNA and tracrRNA are combined into a single RNA cell known as sgRNA, or single-guide RNA, which is then used in DNA targeting.

## 4. Molecular Mechanisms of iPSC Induction in Chickens

The four factors (Oct4, Sox2, Klf4, and c-Myc (OSKM)) were introduced into mouse fibroblasts by Takahashi and Yamanaka in 2006 using retroviral vectors. They succeeded in creating pluripotent stem cells, also known as iPSCs, which resemble ESCs. It would be ideal to stimulate somatic cells to become iPSCs and then further differentiate them into PGCs to get PGCs. There is little information available on somatic cell reprogramming in birds. By using the Nanog, Oct4, Sox2, Klf4, c-Myc, and Lin28 (OSKMNL) lentiviral overexpression vectors, Lu et al. [59] were able to successfully convert somatic cells from quails into iPSCs for the first time in 2012.

### 4.1. Key Factors Involved in Reprogramming

Chicken iPSCs are reprogrammed under various important conditions that improve process stability and efficiency. Initiation and maintenance of the pluripotent state in chicken embryonic fibroblasts depend on core pluripotent factors including Oct4, Sox2, Nanog, and Lin28 (OSNL). Still, the reprogramming system based on these elements by itself usually results in low induction efficiency and instability, demanding additional optimization [60].

The activation of glycolysis leads to a notable improvement in the efficiency of reprogramming. The transition from oxidative phosphorylation (OXPHOS) to glycolysis is a characteristic feature of the reprogramming process. Genes involved in glycolysis, including those responsible for glucose transporters and glycolytic enzymes, are activated during the first phase of reprogramming. Activation therefore increases the production of ATP but reduces that of reactive oxygen species. This metabolic remodeling process is hence crucial in preventing damage to the cell and helps with the fast rate of cell division that is required to maintain pluripotency [61]. Further optimization of the reprogramming efficiency is possible using small-molecule inhibitors, particularly TGF-β and MEK/ERK inhibitors, known as “glycolysis activators”. These molecules act by enhancing the activity of endogenous pluripotency genes with the capacity to differentiate into multiple cell lineages and increasing the metabolic activity associated with glycolysis. Furthermore, the introduction of epigenetic modifiers has been shown to repress exogenous transcription factors while still retaining the inherent ability of chicken iPSCs to give rise to several cell types, thus making them pluripotent. This would work in other species and have high potential for chicken iPSC development [62].

The ability of the mitochondria to function also maintains the features of pluripotent stem cells. Reactive oxygen species are reduced, and the reprogramming environment is enhanced when proteins, such as UCP2, aid in the switch from glucose oxidation to glycolysis. Hypoxia-inducible factor 1 enables the required metabolic transition for reprogramming by serving as a key modulator of the transcription of glycolytic genes [63].

A study conducted in Yamanaka’s lab in 2006 showed that terminally differentiated cells could be reprogrammed into iPSCs by the introduction of four reprogramming factors: Oct3/4, Klf4, Sox2, and c-Myc [64]. Any type of organ, including germ cells, can be differentiated from these iPSCs. In this process, mouse fibroblasts are changed into cells that bear some resemblance to embryonic stem cells with four transcription factors: OCT4, SOX2, KLF4, and c-MYC. Such reprogrammed cells become induced pluripotent stem cells, or iPSCs [65]. NANOG is another crucial transcription inhibitor that prevents a cell’s attempt toward the trophectoderm and extraembryonic endoderm directions [66]. NANOG directly inhibits SMAD Family Member 1 and thus represses bone morphogenetic protein-driven mesoderm growth. NANOG binds to the OCT4 promoter and activates it; thus, NANOG constitutes the important OCT4 transcriptional activator [67]. OCT4, which is necessary for the development of naïve epiblasts, is absent from the inner cell mass of embryos that are OCT4-deficient [68]. Furthermore, the inner cell mass develops trophoblast cells when OCT4 expression in embryonic stem cells is stopped [69]. In both the process of converting adult cells into iPSCs and preserving pluripotency in ESCs, OCT4 plays a crucial role. To regulate the expression of many genes necessary for embryonic development, SOX2 and OCT4 build a complex that absorbs DNA [70].

The reprogramming factors typically come together to create protein complexes, which also establish an interconnected regulatory system. Within reprogrammed cells, this network interacts with additional pluripotency factors, influencing the activation or suppression of numerous genes. Ultimately, it guides the cells towards a genuine state of pluripotency. Within this network, Oct4 and Sox2 serve as crucial elements [71]. Reprogramming takes place in two distinct stages. First, Oct4, Sox2, Klf4, and c-Myc (OSKM) attach to specific areas of chromatin that are inaccessible to other factors found in somatic cells. This leads to chromatin area remodeling and controls the activation or deactivation of many genes. Many chromosomal loci, including those that are not used as locations in embryonic cells where these factors bind, contain OSKM. C-MYC interacts with regions of the genome exhibiting methylation of H3K4, a well-recognized marker of open chromatin. By binding to enhancer and promoter areas that establish the identity of somatic cells, OSKM aids in the regulation of somatic genes. To increase the expression of pluripotency genes, OSKM molecules bind to both their promoters and enhancers at the same time [1].

### 4.2. Signaling Pathways and Gene Expression Profiles

The process of generating iPSCs in chickens requires complex signaling networks and precise gene expression profiles that are crucial for effective reprogramming. The main signaling pathways are the Wnt, TGF-β, MEK/ERK, and HIF1 pathways [72]. One example of a signaling system that encourages mESC growth is Wnt signaling. Glycoproteins that regulate embryonic development are encoded by the Wnt gene family [73]. Important pluripotency genes that are active when the Wnt pathway is engaged include Oct4, Sox2, and Nanog, which are crucial for maintaining pluripotency and are expressed more frequently [74]. B-catenin travels to the nucleus during activation of Wnt/b-catenin signaling, where T-cell factor (TCF) interaction occurs, and lymphoid enhancer-binding factor (LEF) families carry out transcriptional actions as shown in Figure 4. According to some research, undifferentiated mESCs are maintained by Wnt/b-catenin signaling activity [75]. Activating Flat colonies were created because of Wnt signaling from embryonic stem cells (ESCs) and decreased expression of pluripotency markers, potentially because of differentiation. PGCs are a particular kind of stem cell that are present in chickens, and their growth and division are controlled by the activation of Wnt signaling. Throughout the history of evolution, the Wnt/b-catenin pathway has not changed. The suppression of the GSK3b protein is what causes it to happen [76]. Research suggests that the pluripotency of human and mouse embryonic stem cells may be maintained via the Wnt pathway. Targeting glycogen synthase kinase-3 (GSK-3), 6-bromodirubin-3-oxime (BIO) is a particular pharmacological inhibitor that may help preserve the undifferentiated state of embryonic stem cells (ES cells), support the production of markers unique to ES cells, and stimulate the Wnt pathway. Wnt signaling normally increases and decreases as embryonic stem cells develop [77].

For embryonic stem cells (ES cells) to develop, survive, and maintain their pluripotency, the PI3K pathway is essential. ES cells specifically express eras, which causes PI3K to become active. PI3K activation encourages ES cell proliferation [78]. In addition to encouraging the growth of ES cells, PI3K activity could also be necessary for the self-renewal of ES cells [79]. Growth factors such as bFGF and LIF that support ES cell pluripotency may activate PI3K/Akt signaling [80]. Additionally, to prevent ES cell death, PI3K/Akt signaling is necessary. When it comes to ES cells self-renewing, density is crucial. Thus, PI3K may also be involved in the process of the self-renewal of ESCs [81].

The factor responsible for promoting growth and inducing differentiation TGF-β is an exemplary member of an extensive superfamily. The family consists of more than 40 members, including bone morphogenetic proteins (BMPs), TGF-β, activin, and nodal. These ligands attach themselves to embryonic stem (ES) cells. The heteromeric complex of serine/threonine kinase receptors known as TGF-β type I and type II receptors interacts to transfer the TGF-β signal from the membrane to the nucleus. Type I receptors do not connect to TGF-activin unless type II receptors are present, even though type II receptors are highly attracted to it [82]. Through heterodimers of TGF-βR 1 and 2, TGF-β ligands such as activin A and nodal phosphorylate regulatory Smad (R-Smad) proteins. In 1–2 h, hundreds of genes are up- or down-regulated because of R-Smads binding to co-Smad (Smad4) after phosphorylation and translocating into the nucleus (Figure 5) [83].

Reprogramming requires the suppression of the TGF-β route to enable the transition from mesenchymal to epithelial cells. LIF and BMP4 play crucial roles in supporting the ability of mESCs to self-renew. STAT3, a transcription factor activated by LIF, is crucial for preserving the capacity of mESCs to self-renew. To properly activate the differentiation of inhibitory (Id) genes and prevent brain differentiation while promoting the self-renewal of mESCs in culture, BMP must be used together with LIF. LIF by itself is not enough to achieve this [84].

ERK1 and ERK2 belong to the family of mitogen-activated protein kinases. They govern basic cellular processes such as motility, differentiation, development, survival, proliferation, and metabolism [85]. While Akt/mTOR signaling has a more significant role in cell proliferation, ERK signaling is also vital. For example, ERK uses several pathways to cause RNA polymerase I (Pol I) to stimulate the transcription of genes for ribosomal DNA, Pol III to trigger the transcription of tRNA genes, and Pol II to trigger the transcription of ribosomal protein genes. Furthermore, ERK stimulates ribosomal protein S6 kinase, mTOR complex 1, and MAPK-interacting kinase, which are activated by ERK as shown in Figure 6 [86]. Prior research has shown that ERK has a role in the production of proteins in the skeletal muscle of chickens. For instance, in chicken myoblasts, an ERK inhibitor eliminates the phosphorylation of S6K1 produced by insulin [87]. When receptors are engaged, the ERK pathway is triggered by the interaction of a complex including the Grb2 adaptor and the Sos guanine nucleotide exchange factor. Raf and MAPK kinases (MEK) initiate a series of transphosphorylations that activate ERK when Sos, a membrane-bound protein, activates Ras [88]. In contrast, inhibiting the MEK/ERK pathway helps to maintain pluripotency by preventing cells from differentiating [89].

Viral vectors play a role in the introduction of the essential pluripotency genes (Oct4, Sox2, Nanog, and Lin28) when reprogramming is taking place. Consequently, there is a discernible rise in the expression of these genes [90]. Additionally, there is an observable elevation in the way that glycolysis-related genes are expressed, such as lactate dehydrogenase A (LDHA) and glucose transporters such as GLUT1. This indicates a change towards glycolytic metabolism, which is essential for iPSCs [91]. The process of reprogramming also includes the activation of different epigenetic modifiers that aid in suppressing foreign transcription factors while preserving native pluripotency genes, thereby ensuring a consistent reprogramming process [92]. In addition, there is a reduction in the expression of mesenchymal markers and an increase in the expression of epithelial markers throughout the MET process. The MEK/ERK and TGF-β signaling pathways are the main factors affecting this [93]. These pathways and gene expression patterns illustrate the complex regulatory mechanisms necessary for pluripotency to develop and be maintained in chicken iPSCs. Reprogramming methods may be made more effective with the use of this knowledge.

Recent research demonstrates that transcription factors Oct4, Sox2, Nanog, and others are involved in the transcriptional-regulatory network that is important for the self-renewal and differentiation of ES cells. Particularly, only pluripotent cells express Oct4 and Nanog. The self-renewal and differentiation of ES cells could be dependent on these transcription factors. The activation of genes is influenced by external signals, such as the ones we recently talked about. Genes also have their own regulatory mechanisms. A transcription factor known as Oct4, also called Oct3 at times, attaches to the octamer sequence ATGCAAAT. Pou5f1 is responsible for its production. Oct4 expression reaches its highest point during the four-cell stage of mouse preimplantation development, and it is only observed in pluripotent cells such as germ cells, ES cells, and ICM. For the maintenance of the ES cells’ current state, it is crucial for this gene to be present in the exact levels required. Overexpression of Oct4 encourages development into primitive endoderm and mesoderm, while the absence of Oct4 causes ES cells to incorrectly differentiate into trophectoderm [94]. Oct4 has been shown to target many genes in ES cells, including Fgf4, Utf1, Opn, Rex1/Zfp42, Fbx15, and Sox2 [95]. The high mobility group (HMG) family of proteins includes a member known as the “HMG-family protein”. Sox2 plays a crucial role in preserving the pluripotent status of embryonic stem cells and shares common gene targets with Oct4. The precise concentration of Oct4 significantly influences the fate of primitive cells. Maintaining Oct4 protein levels within a specific range is crucial for sustaining the pluripotent state. Oct4’s expression may be self-regulated [96].

The transcription factor Nanog, which contains homeobox, is essential for preserving the embryonic stem cells and pluripotent cells found in the inner cell mass. Pluripotent cells have it, but mature cells do not. When ES cells have their Nanog disrupted, these cells differentiate into endoderm lineages. Overexpressing Nanog in mouse ES cells allows the cells to proliferate independently without needing LIF. The cells’ ability to self-renew is decreased, indicating that Nanog plays a crucial role in the regulation of the pluripotent state [96]. Nanog mainly acts as a transcriptional suppressor for downstream genes such as Gata4 and Gata6, which are essential for preserving pluripotency and participating in cell differentiation [96]. Additionally, various studies, such as RNAi-mediated knockdowns, in vitro binding experiments, Chip analysis, and other data, have shown that the Oct4/Sox2 complex directly targets the promoter [96].

Sox2 is a transcription factor that is a member of the Sox family. The first characterization of the Sox gene family was provided by the identification of Sry, the mammalian testis-determining factor [97]. The DNA-binding high-mobility-group (HMG) domain is a highly conserved feature shared by all Sox family members. Sox2, Sox1, and Sox3 are classified as members of the SoxB1 group. Sox1, Sox2, and Sox3 have comparable roles and a sequence that is around 80% identical. However, Sox2 is critical for the development of embryos and performs a variety of context-specific functions. Various variables have been shown to impact the Sox protein’s binding to its target genes, resulting in a wide range of functional consequences [97].

Sox2 significantly influences the early growth of pluripotent embryonic cells. The lack of Sox2 does not hinder the creation of trophectoderm. Nevertheless, embryonic death occurs when Sox2 is deleted in the zygote because of the incapacity to produce pluripotent epiblast [98]. Sox2 plays an essential role as a transcription factor, essential for maintaining the ability of stem cells to differentiate into different cell types while preserving their pluripotency. When Sox2 and Oct4 proteins come together, they form a binary complex that can attract other nuclear factors. These genes are connected to repression of differentiation-related genes and activation of pluripotency-related genes. Moreover, Sox2 plays a critical function in initiating the neural induction process and maintaining the characteristics of neural progenitor stem cells throughout neural differentiation. It has been recently found that Sox2, although expressed in adult stem cells residing in several epithelial tissues, is required for the regulation of myeloid stem cell growth [99]. Thus, Sox2 has a huge contribution to generating pluripotent cells; it also plays an essential role in the embryo during its early days of development.

The c-myc protein is associated with numerous functional domains, such as a DNA-binding basic region, a helix-loop-helix, a leucine zipper domain, a trans-activational domain composed of Myc box I and Myc box II, and NLS. The transcription factors c-myc1 and c-myc2 have been found to be either activators or repressors of target gene transcription. This likely happens owing to their possessing both the MbI and MbII domains involved in transactivation and trans-suppression, respectively. In contrast, lacking the MbI domain has shown that c-mycS acts as a transcription repressor [96]. The protein c-myc takes part in various physiological processes required for the proper regulation of the cell cycle, differentiation, and apoptosis. Facchini et al. have reported that at the beginning, it either activates, represses, or does both, to several c-myc target genes during the initiation of these processes. In normal cells, c-myc expression is strictly regulated to provide controlled proliferation. To control cell cycle progression and track cellular development, normal cells must both transcriptionally activate growth stimulatory genes and transcriptionally decrease growth inhibitory genes of c-myc target genes [100].

## 5. Efficiency and Challenges of Different Techniques

The process of converting chicken somatic cells into induced pluripotent stem cells (iPSCs) encompasses a range of approaches, each characterized by unique levels of effectiveness and difficulties. The process of reprogramming utilizing viral vectors, specifically retroviral and lentiviral vectors, is extremely effective, with reprogramming rates ranging from 0.1% to 1%. Nevertheless, this approach carries substantial hazards because of the incorporation of viral vectors into the genetic material of the host, which can result in the introduction of mutations and the potential development of tumors. The utilization of integrating viral vectors is restricted due to the safety risks they provide, particularly in clinical settings [101].

Non-integrating viral techniques, such as the Sendai virus, provide a safer option with a moderate to high level of effectiveness, such as integrating viral vectors, which usually range from 0.1% to 0.5% [102]. The primary difficulty associated with Sendai virus is its temporary expression, which requires many transductions to sustain the necessary amounts of reprogramming factors for the successful creation of induced pluripotent stem cells (iPSCs). This introduces intricacy to the process of reprogramming [103]. Another non-integrating method is the use of episomal plasmids, which typically have a reprogramming efficacy of 0.01% to 0.1% [104]. Cell division eventually results in the loss of these plasmids; therefore, effective cell reprogramming and expansion are necessary. Because there is no genomic integration, the approach is safer, but because of its reduced effectiveness, transfection settings must be optimized to make it more feasible [105].

Unlike viral approaches, the PiggyBac transposon system exhibits a high reprogramming efficiency, typically ranging from 0.1% to 1% [106]. With this technique, transposons containing reprogramming factors are integrated into the host genome, but they can be removed later, leaving no trace of their presence [107]. Even though the initial integration is stable, it can be difficult to ensure that the transposon is completely removed without leaving any residue, necessitating extra steps for accurate excision [108].

Protein-based reprogramming, which involves the direct administration of recombinant proteins containing reprogramming factors, eliminates genetic changes but has low efficiency [109]. The primary obstacle is delivering proteins into cells efficiently because of their short half-lives, which require repeated delivery and provide major technical challenges [110]. Using the CRISPR/dCas9 system, CRISPR-based activation (CRISPRa) provides a precise method for upregulating endogenous reprogramming genes [111]. However, depending on how CRISPR/dCas9 components are delivered and expressed, its effectiveness varies and is frequently lower than that of conventional viral techniques. To accomplish reliable reprogramming, the method’s technical complexity and potential off-target consequences necessitate careful optimization [112].

To summarize, while some reprogramming methods are highly efficient, they also present substantial obstacles, notably in the form of safety and in technical practices. The approach used is determined by the individual study requirements as well as the desired balance of efficiency and safety.

## 6. Genetic and Epigenetic Considerations

Viral vectors merge into the host DNA, raising the possibility of insertional mutagenesis, which can damage genetic integrity and lead to undesired genetic changes as shown in Table 1. They may also produce abnormal epigenetic changes [113]. Unlike viral vectors, the Sendai virus does not combine with the DNA of the host, resulting in minimal genetic damage. However, due to its temporary nature, repeated transduction may still offer certain epigenetic dangers [114]. Episomal plasmids do not enter the host genome, resulting in substantially less genetic damage. However, they may be lost over time, necessitating multiple transfections to sustain gene expression [115]. The PiggyBac Interchange System integrates at genomic locations and has the capacity to modify genetic information precisely. To prevent lingering genetic alterations, the removal of these components might be a difficult procedure that needs to be carefully controlled [116]. Protein-based reprogramming prevents genomic changes by delivering reprogramming factors directly as proteins. However, these proteins can be difficult to transport and stabilize, which may have an impact on epigenetic states [117]. Stem-based CRISPR activation allows for precise targeting of genetic activation, yet there is a risk of unintended genetic and epigenetic modifications due to off-target effects. Optimizing delivery is essential for reducing these dangers Figure 7 [118].

## 7. Applications of Chicken iPSCs

Chicken iPSCs have become an instrumental model in both fundamental and practical scientific investigations. These cells have been seen as a valuable model in research areas associated with developmental biology, gene function, and the mechanism that induces cellular differentiation, as shown in Figure 8. In the discipline of agricultural science, chicken iPSCs are bound to have the most impact on genetic improvement. The strengths include improvements in disease resistance, growth rate, and meat quality [120]. They are also important to the field of regenerative medicine and may have applications in tissue engineering and the creation of cell-based therapeutics for birds. In addition, chicken iPSCs are an invaluable tool to conduct virology research and vaccine development, especially the study of poultry diseases such as avian influenza [121].

### 7.1. Regenerative Medicine and Tissue Engineering

These stem cells can become any cell type and have been obtained from chickens, showing vast potential for tissue engineering and regenerative medicine. They are well-suited for developing bird models that investigate tissue regeneration and the healing process by creating various cell types [123]. Thus, such iPSCs from chickens will be of invaluable use in the assessment and development of new treatments against avian diseases, as they can reproduce tissues and organs through in vitro culture and bioengineer complex tissues for transplantation, which may transform veterinary medicine by providing personalization to regenerate therapy in poultry. In 1961, Till and McCulloch made a groundbreaking discovery: every living cell can renew itself. They conducted an experiment where lethal radiation dosages were given to mice and then injected with bone marrow cells. The researchers observed that the survival of the mice depended on the clusters of cells produced by cloned mouse cells [124]. Regenerative medicine may use iPSCs to replace damaged or deteriorated tissues. These cells are cultured in a lab before being implanted at the precise site of degeneration or damage. Gene therapy is faced with significant obstacles concerning the availability of organs or tissues and immunorejection. The unavailability of organ donors and the growing need for organs due to degenerative diseases and accidents often result in people not being able to find suitable donors, leading to loss of lives. Additionally, patients can only receive organ, tissue, or cell transplants from donors who are healthy and have a compatible physiological makeup. Before implanting tissues or organs into a patient’s body, several tests are carried out to consider these potential risks. Using iPSCs presents a promising approach to these kinds of therapies since the transplanted cells are regenerated iPSCs that come from somatic cells of the patients. iPSCs have been used to treat a range of degenerative illnesses and injuries [53]. To rectify the genetic deficit, Kazuki et al. used iPSCs generated from a patient suffering from Duchenne muscular dystrophy (DMD). To produce the whole dystrophin (DYS) sequence, they used a human artificial chromosome (HAC). Duchenne muscular dystrophy-containing induced pluripotent stem cells (iPSCs) were created using the patient’s fibroblasts. The dystrophin genomic sequence was transferred in its entirety into a DYS-HAC (Human Artificial Chromosome) using microcell-mediated chromosomal transfer (MMCT), resulting in the repair of the dystrophin gene in iPSCs that had been deleted or mutated [125]. Currently, there is a growing body of research about the use of iPSCs for the ex vivo growth of different blood components. They may be employed to generate red blood cells (RBCs) that are needed worldwide for the treatment of different injuries or disorders. There are many methodologies available for using ESCs/iPSCs in the generation of red blood cells (RBCs) [126]. Degenerative diseases characterized by cell death and the subsequent onset of various symptoms have been addressed through gene therapy utilizing iPSCs. Another such disease is retinitis pigmentosa, more commonly referred to as retinal degeneration, which causes reduced vision within the eye. In the instance of RP treatment, iPSCs of the RP patient were generated and shown to have expressed rod photoreceptor cells [127]. Application of chicken-induced pluripotent stem cells in these areas can provide new opportunities not only to improve poultry welfare and productivity but also to advance our knowledge regarding stem cell biology and the principles of tissue engineering [128].

### 7.2. Genetic Engineering and Transgenics

Chicken iPSCs are completely changing the face of genetic engineering and transgenics. The cells can be easily modified to add, edit, or eliminate genes that can be used in making transgenic chickens that will have improved resistance to diseases, fast growth, high-quality meat, among other characteristics. Chicken-induced pluripotent stem cells are permissive for the introduction of precise genetic modifications, a step that has been deemed a requirement to study the function and regulation of genes in bird species [129]. Second, they represent an ideal vehicle for the expression of recombinant proteins and biopharmaceuticals within the context of a system that is easily and inexpensively grown to produce often complex biological products, as shown in Figure 9. In the late 1980s, gene therapy launched in the United States under the initiative of oncologist Steven Rosenberg. A retroviral vector containing a genetic marker was used to trace the T cells that were reinfused. Genetic engineering and transgenics [130]. Rogers and Pfuderer’s research [60] was the first to show that viral RNA/DNA can be used to transmit genetic material via transduction. In 2003 [61], Genidicine, the initial gene therapy drug, was approved in China. Genedicine has minimal side effects when utilized for adenoviral therapy in squamous cell carcinoma. The approval of this medication expanded the field of genetic therapy. LPL deficiency is treated by restoring lipoprotein lipase expression using Gylbera, an adenoviral vector that is the first gene therapy medication approved for commercial use in Europe [131]. The CRISPR-Cas9 method was used on an iPSC model of retinal degenerative diseases recently. Leber congenital amaurosis (LCA) is an uncommon hereditary disorder that causes damage to children’s retinas. A specific alteration in the CEP290 gene occurs in the non-coding region called an intron, leading to the most severe form of LCA [132]. Researchers have successfully eliminated the mutation in patient-derived iPSCs using the CRISPR-Cas9 gene editing technique [133]. The inability of the vector to accommodate the entire length of the cDNA has ruled out AAV-mediated gene therapy for this type. The advancements in genetic engineering using chicken iPSCs are paving the way for significant improvements in poultry breeding programs, animal welfare, and the production of high-value agricultural products [134].

### 7.3. Disease Modeling and Drug Testing

There has been a lot of interest in using chicken cells to create iPSCs for drug testing and studying diseases [136]. ciPSCs offer an advanced platform for researching diseases that impact chickens, a major source of protein globally, and avian biology [137]. When it comes to disease modeling, reprogramming ciPSCs can generate differentiated cell types that accurately replicate the diseased state [138]. Using iPSC technology, chemicals derived from pathological research are now being utilized in clinical trials, with a widening scope of target disorders (Figure 10) [139]. This includes conditions such as Pendred syndrome (PDS), fibrodysplasia ossificans progressive (FOP), and ALS [140]. Researchers found that sirolimus, a mTOR inhibitor, may be used to treat cochlear cells in PDS patients using an iPSC model. Following the completion of a phase I/IIa double-blind parallel-group single-center trial for PDS patients, scientists may now investigate disease processes at the microscopic level [62]. Moreover, iPSC technology, there has been a growing interest in creating induced neurons (iNs) through alternate reprogramming methods. Somatic cells, such as skin fibroblasts, can be directly transformed into induced neurons (iNs) by introducing transcriptional factors [63] and microRNAs (e.g., miR9/124) into them. This technique directly converts original cells into neurons, bypassing the intermediate induced pluripotent stem cell (iPSC) stage. Utilizing induced neurons (iNs) could significantly aid research on neurodegenerative diseases that manifest later in life, as they retain the aging characteristics of fibroblasts, including the epigenetic clock [64]. For example, ciPSCs have been utilized to simulate viral infections in chickens, offering valuable understanding of the interactions between the host and the virus and facilitating the creation of antiviral approaches. Therefore, CiPSCs could be used to offer a renewable source of specific cells for the screening of potential medicinal compounds [141]. This technology will go a long way towards benefiting the poultry sector by providing effective and cheaper means for drug testing. These could further lead to better treatments for diseases that might affect chicken health and production. Some studies have harnessed ciPSCs for in vitro testing of antiviral drugs against avian influenza, which illustrates the utility of these cells in the development of effective therapies and the reduction of live animal experimentation [65].

### 7.4. Agricultural and Animal Production

iPSCs have huge potential use in agriculture and veterinary science, of which impacts have already been felt in the poultry industry. Cellular reprogramming of iPSCs could be used to maximize the result of genetic breeding activities by facilitating precise genetic editing [66]. This ability enables researchers to genotype favorable traits—for example, resistance to diseases, growth rates, improved feed utilization, or any other characteristics being sought after—into chickens. For instance, iPSCs have been used in the production of transgenic chickens that are more resistant to avian diseases. This has provided improved health for the entire flock and minimized the use of drugs [67]. Diseases such as PRRS in pigs, which are important in agriculture, need the implementation of iPSCs to confer resistance. The pork industry suffers a great deal of economic loss due to the highly infectious viral disease known as PRRS. Scientists have used iPSCs to modify the genetics of pigs by focusing on the CD163 gene. This forms a basis that greatly enhances the entrance of the virus into the cell. Scientists have therefore been successful in modifying this gene to produce pigs that are resistant to PRRS [143].

The ability of iPSCs to help endangered animal species and conservation, to restore extinct species, and to reduce consumers dependence on animals will be deeply valued. Induced pluripotent stem cells (iPSCs) could not be differentiated from ESCs, had the property to proliferate indefinitely, and possessed the potential to give rise to all three primary germ layers. Unlike ESCs, the generation of iPSCs does not require the acquisition of embryonic tissues or oocytes. Embryos from endangered species are usually not available in sufficient numbers; induced pluripotent stem cells generated from somatic tissue are a more pragmatic source of stem cells, and fewer moral and ethical concerns surround their use. It also offers significant advantages for the use of iPSCs from domestic animals in reducing animal mortality associated with the commercial production of animal products [144]. Domesticated livestock deplete a large number of natural resources through deforestation for grazing, require extensive water use, and generate immense greenhouse gas emissions. The alternative animal products made from the iPSCs could help reduce environmental damages brought by intensive farming and have sustainable economic uses. The application of iPSCs derived from domestic animals, such as cattle and swine, to produce clean meat could reduce the environmental impact caused by conventional animal husbandry. Applications of iPSCs to obtain rare animal products without harming the animals are also considered [145].

iPSCs are a new means for generating regenerative medicines and vaccinations in veterinary medicine. In vitro models generated by iPSCs’ differentiation into various lineages can be used to explore avian diseases, therefore helping in the final development of more efficient and directed therapeutic solutions. For example, ciPSCs have been differentiated into immune cells to better comprehend the avian immune response. Such technology has made it easier to develop vaccinations that provoke a more potent and specific immune response [146]. Further, chicken congenital abnormalities or injuries could be repaired by means of tissue grafts enabled by the ciPSCs, improving the welfare and productivity of the animals [147].

### 7.5. Treatment of Virus-Related Pandemics

The use of iPSC technology to combat pandemics caused by viruses holds great promise. Recent progress in biotechnology regarding the reprogramming of cells and generating iPSCs revolutionized methodologies used for studies into the mechanisms of human disease and testing new pharmaceutical agents. This technology can also be applied to produce patient-specific models for investigations of the host–pathogen relationship and for developing novel anti-microbial and anti-viral therapies. Some applications of iPSC technology to investigate viral infections in humans including the modeling of human genetic predispositions to serious viral diseases that include encephalitis and serious influenza and genetic engineering and genome editing in iPSC-derived cells obtained directly from a patient to introduce antiviral resistance [148].

The ability to reprogram human-induced pluripotent stem cells from donors’ somatic cells has opened new avenues for the study and understanding of the basic pathophysiology of human diseases, including a growing list of viral infections such as Zika virus (ZIKV), hepatitis C virus (HCV), and influenza virus (H1N1). Cellular reprogramming toward iPSC generation is highly cumbersome due to high costs, the requirement of a great amount of resources, and the tendency of iPSCs to revert to their original somatic genotypes over time. The restricted availability of donor cells remains a formidable obstacle, especially in the creation of new pharmacological interventions against viral and other infectious diseases [149]. The first in vitro study that mimicked ZIKV infection in human brain cells employed two-dimensional cultures of iPSC-derived neural progenitor cells and neurons [41]. That work identified that the ZIKV mainly infected neural progenitor cells, and not the induced pluripotent stem cells or neurons, causing increased cell death and interference in the regular course of the cell cycle. Another study independently showed that ZIKV infects human neural crest cells and peripheral neurons derived from stem cells in vitro and causes increased cell death and transcriptional dysregulation [42]. These examples demonstrate the application of 2D human stem cell-derived models to the study of viral tropism and cell type-specific pathogenesis [150].

Cell lines derived from induced pluripotent stem cells specific to individual patients allow the personalization of drugs, improving the therapeutic effectiveness. Bietti crystalline dystrophy is a rare disorder of visual impairment, and it is expected to affect about 67,000 persons worldwide [151]. In cell treatments, iPSCs have emerged as a transforming resource owing not only to their capacity for development into many cell types but also due to an inexhaustible supply and the promise of readily available cell products. Recent advancements in iPSC-derived immune cells have produced powerful natural killer (iNK) and iT cells that, in animal models and clinical studies, displayed significant efficacy in eliminating cancerous cells [152].

## 8. Challenges and Future Directions

Reprogramming methods such as viruses and integrative techniques are often the most effective, but they present the lowest degree of safety due to possible hazards such as insertional mutagenesis and the presence of viral components. Nonetheless, using less risky techniques such as tiny chemicals (such as tranylcypromine (first introduced by Smith, Kline and French in the United Kingdom in 1960), valproic acid (first synthesized in 1881 by Beverly S. Burton), and RepSox (discovered by a team at the Harvard Stem Cell Institute. They found a molecule that could replace the Sox2 gene in reprogramming cells) is an option. iPSCs offer multilineage differentiation capacity and immortality, making them the perfect source of ESCs for any biological application [153]. MicroRNAs (such as the miR-302~367 cluster, miR-371~373 cluster, and miR-17 family) [109] or metabolites (such as sodium butyrate, ascorbic acid, and forskolin) [154] possess reduced reprogramming efficacy or are usually unable to induce pluripotency on their own [23]. As a result, they are frequently used together with traditional reprogramming factors [32]. Introducing mRNAs that encode the typical reprogramming factors can induce pluripotency in somatic cells. Nevertheless, the mRNAs require multiple transfections because of their inherent instability. Reverse transcription can convert the mRNAs into DNA and integrate them into transfected cells’ genomes, making this approach labor-intensive and posing additional risks. Genomic integration can result in the disturbance of tumor suppressor genes and/or the abnormal and irreversible activation of proto-oncogenes. This can potentially cause genetically modified cells to undergo malignant transformation.

The in vitro creation, growth, and differentiation of iPSCs may result in detrimental epigenetic aberrations and/or genetic alterations, which might be artificially induced by culture adaptation [155]. Aberrant epigenomic or genomic alterations might impact the proliferation, differentiation, and functionality of iPSC lines, hence affecting their usefulness for subsequent applications [23]. Thus, it is preferable to utilize reprogramming (and differentiation) procedures that have rapid kinetics to reduce culture-induced epigenetic and genetic alterations. According to a study by Lu et al. (2020), variations in gene regulation and cellular mechanisms between species can be the reason for the reduced reprogramming effectiveness of chicken somatic cells in comparison to mammalian cells. Chicken-specific reprogramming factors must be identified and optimized because the conventional reprogramming factors (Oct4, Sox2, Klf4, and c-Myc) utilized for mammalian cells are frequently less effective in avian cells. Furthermore, the requirement for growth circumstances, such as the creation of media, substrates, and feeder layers that support pluripotency, makes it difficult to sustain chicken iPSCs in culture. The use of chicken iPSCs is also impeded by genetic and epigenetic instability in reprogramming and long-term culture, which may affect their potential for differentiation into these cells and their in vivo functionality.

Future efforts should be focused on the optimization of procedures and the improvement of reprogramming efficiency by discovering chicken-specific reprogramming factors. Sophisticated culture methods are required to maintain the pluripotency and genetic stability of chicken iPSCs. For better reprogramming techniques, an understanding of the biochemical and genetic principles that drive pluripotency and differentiation in chickens will be required. Ethical frameworks and regulatory guidelines should be set up to make sure that responsible and humane methods will be used when iPSC technology is used on chickens. Looking at how chicken iPSCs could play a part in biotechnology and agricultural applications, taking into consideration the concerns articulated by society, we can realize big strides in these areas; what matters is long-term feasibility.

## 9. Conclusions

Avian biotechnology research relating to induced pluripotent stem cells has developed considerably in the last decade due to increasing knowledge and utilization of this revolutionary technology. Owed to the great economic and scientific importance of the chicken model, it has been established as representative of the avian species and a center of focus in the studies of induced pluripotent stem cells. Various challenges have been encountered in the establishment of pluripotency in chicken cells, which include low reprogramming efficiency and optimization of the reprogramming factors with species specificity, besides the requirement to continue working under growth conditions that should be continuously identical. There are abundant ethical issues that call for humane standards to be implemented by a well-designed regulatory framework, one that can ensure responsibility in progress.

However, chicken iPSCs provide the flexibility to a wide range of scientific investigations, right from basic biological research, disease modeling, and biotechnological advances to agricultural benefits. Hence, with these advances made on reprogramming methodologies and understanding the basic molecular routes up to the ground level in optimization, many efforts have been made on which future investigations are directed to optimize the efficiency and robustness of chicken iPSCs.

Although these studies indicate some serious technical difficulties, chicken iPSCs still have huge potential for a very diversified set of applications. It provides a flexible platform that covers very broad spectra of scientific studies and real-life applications, from basic biological research and modeling of human diseases to biotechnological improvements and agricultural applications. Such progress in the optimization of reprogramming methods and an understanding of the central molecular pathways laid firm ground for further studies aiming at enhancing the efficiency and stability of chicken iPSCs.

## Figures and Tables

**Figure 3 vetsci-11-00666-f003:**
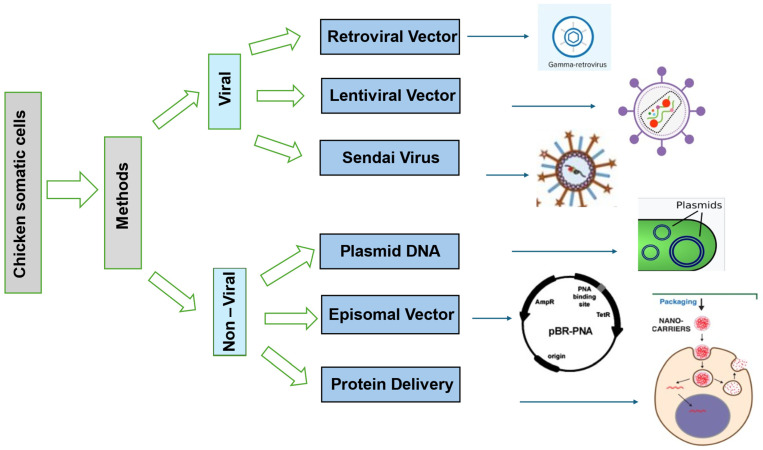
Methods of Reprogramming Chicken iPSCs [33,34,35,36,37,38].

**Figure 4 vetsci-11-00666-f004:**
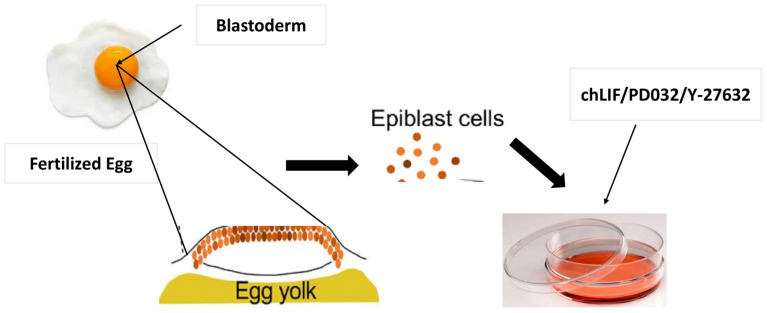
Wnt Pathway Mechanism [76] reprinted and modified with permission from [76]. Copyright 2024, Elsevier.

**Figure 5 vetsci-11-00666-f005:**
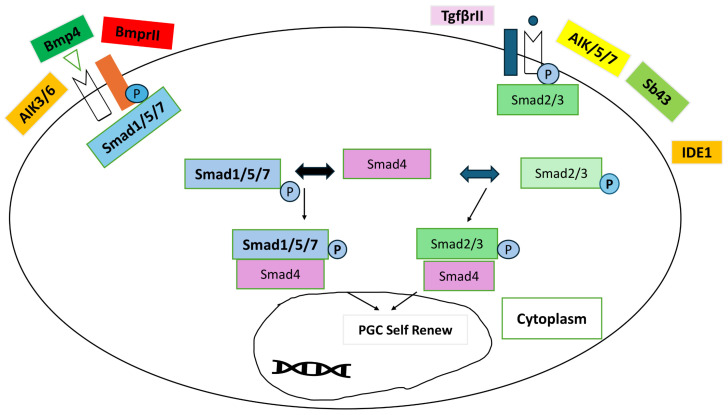
TGF-β/activin/nodal signaling pathway mechanism [82].

**Figure 6 vetsci-11-00666-f006:**
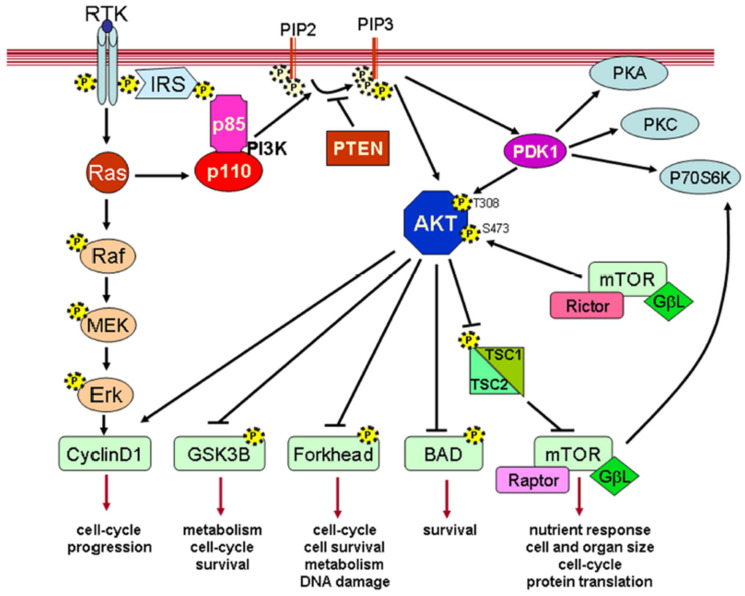
Mechanism of ERK/MPK signaling pathway [88].

**Figure 7 vetsci-11-00666-f007:**
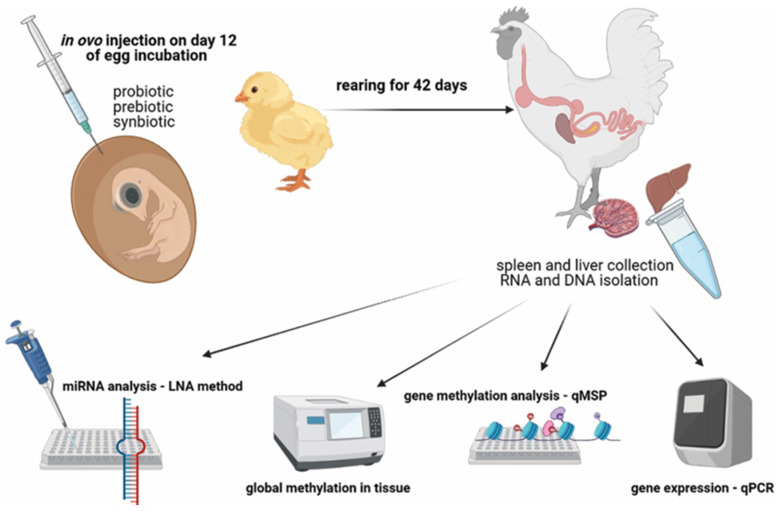
Maintenance of Epigenetic Reprogramming technique [119].

**Figure 8 vetsci-11-00666-f008:**
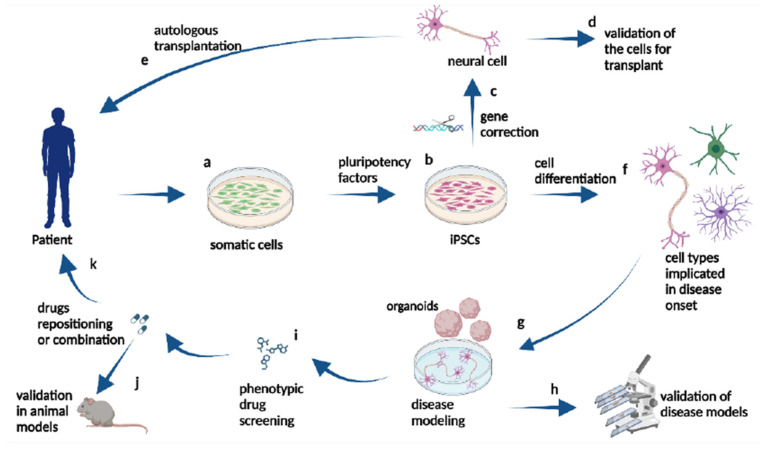
Uses of iPSCs in different fields [122].

**Figure 9 vetsci-11-00666-f009:**
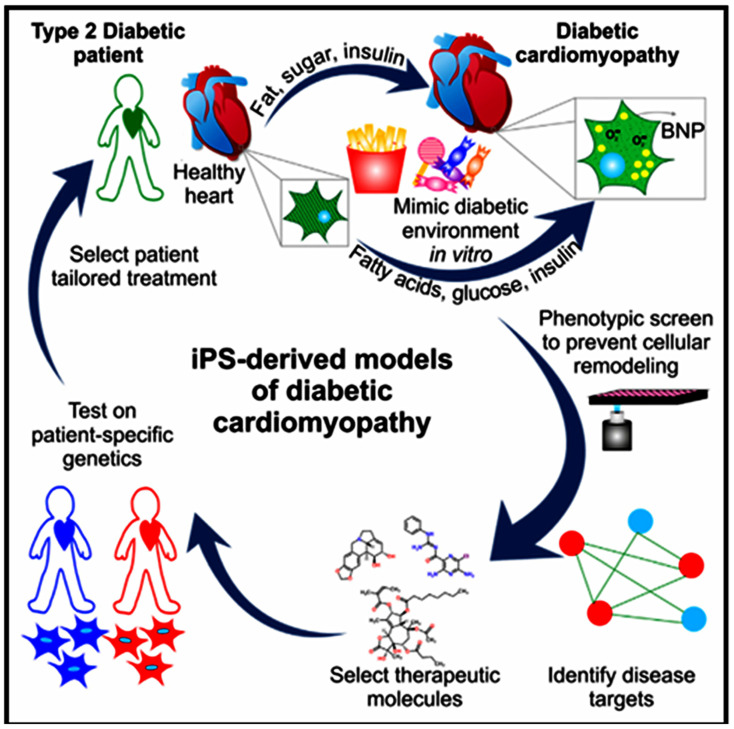
Disease modeling and drug testing [135] reprinted with permission from [135]. Copyright 2014, Elsevier.

**Figure 10 vetsci-11-00666-f010:**
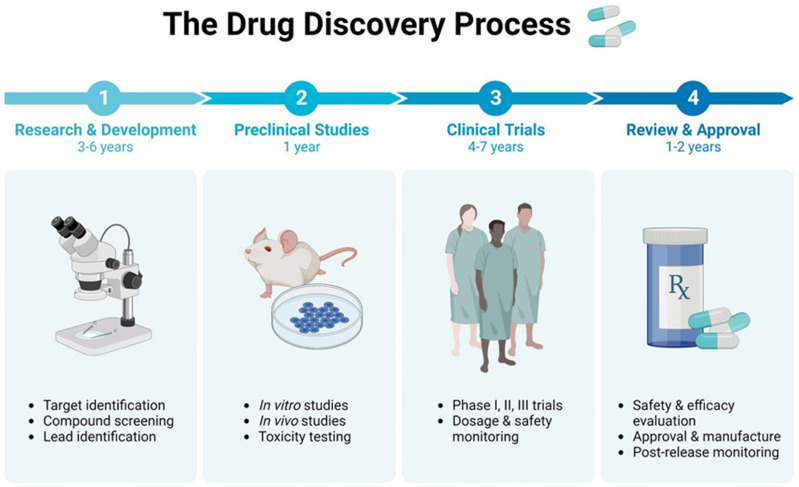
Process of drug testing [142].

**Table 1 vetsci-11-00666-t001:** Genetic and Epigenetic considerations.

Vectors	Genetic Considerations	Epigenetic Considerations
**Viral Vectors**	Integration into host genome; risk of insertional mutagenesis	May cause aberrant epigenetic changes due to random integration
**Sendai Virus**	Does not integrate into host genome; transient expression	Minimal epigenetic impact; requires repeated transduction for efficiency
**Episomal Plasmids**	No integration; extrachromosomal; may be lost over time	Reduced risk of epigenetic alterations; may require repeated transfection
**PiggyBac Transposon System**	Integration into specific sites; excision complexity	Potential for precise epigenetic reprogramming; requires careful monitoring for complete removal
**CRISPR-Based Activation**	Precision targeting; potential off-target effects	Can induce specific epigenetic changes; delivery optimization required

## Data Availability

No new data were created in this paper.

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
