# Peer review of "Induced Pluripotent Stem Cells in Birds: Opportunities and Challenges for Science and Agriculture"

_vetsci, 2024, doi:10.3390/vetsci11120666_

Round 1

Reviewer 1 Report

Comments and Suggestions for Authors

I appreciate the opportunity to review the manuscript entitled “Advancements and Applications of Induced Pluripotent Stem Cells (iPSCs) in Avian Models: Insights from Chicken Studies.” The manuscript is very interesting; however, I recommend addressing the following points:

  • Clearly state the source of the figures or provide confirmation of permission to use the images.
  • Improve the quality of Figure 10, as adjustments are needed to enhance its clarity and resolution.

Author Response

I appreciate the opportunity to review the manuscript entitled “Advancements and Applications of Induced Pluripotent Stem Cells (iPSCs) in Avian Models: Insights from Chicken Studies.” The manuscript is very interesting; however, I recommend addressing the following points:

Comment 1: Clearly state the source of the figures or provide confirmation of permission to use the images.

Response:

  • Figure 1. Process of Reprogramming iPSCs 19 reprinted and modified with permission from ref. 19. Copyright 2016, Elsevier.

    Figure 2. Historical background of iPSC production self-prepared

    Figure 3. Methods of Reprogramming Chicken iPSCs

    Figure 4. Wnt Pathway Mechanism 76 reprinted and modified with permission from ref. 76. Copyright 2024, Elsevier.

    Figure 5. TGF-β/Activin/Nodal Signaling Pathway Mechanism82 reprinted and modified from an open access journal from ref. 76. Copyright 2023, Iranian Journal of Applied Animal Science.

    Figure 6. Mechanism of ERK/MPK Signaling pathway88 reprinted from an open access journal from ref. 88. Copyright 2014, Frontiers.

    Figure 7. Maintenance of Epigenetic Reprogramming technique119 reprinted from an open access journal from ref. 119. Copyright 2021, Animal Frontiers.

    Figure 8. Uses of iPSCs in different fields122 reprinted from an open access journal from ref. 122. Copyright 2024, International journal of molecular sciences.

    Figure 9. Disease Modeling and Drug testing135 reprinted with permission from ref. 135. Copyright 2014, Elsevier.

    Figure 10. Process of Drug Testing142 reprinted from an open access journal from ref. 142. Copyright 2023, Cellular & Molecular Biology Letters.

    Comment 2: 
  • Improve the quality of Figure 10, as adjustments are needed to enhance its clarity and resolution.
  • Response:
  • i have change the figure 10 because i found this figure more suitable for this text.

Reviewer 2 Report

Comments and Suggestions for Authors

After reviewing the manuscript entitled "Advancements and Applications of Induced Pluripotent Stem Cells (iPSCs) in Avian Models: Insights from Chicken Studies" by Zahoor et al. submitted for consideration as a potential publication in the journal Veterinary Sciences, I would like to express the following as a reviewer:

  1. The authors present a detailed and thorough review of the role of iPSCs in avian models, including their definition, history, induction mechanisms, and potential clinical and zootechnical applications, as well as their translational value. This review contributes to the existing literature by summarizing the most relevant studies and, based on this information, opens new avenues for the use of these cells in avian models and their potential applications in humans and animals.
  2. I suggest that the title of the manuscript be changed to the following: "Induced pluripotent stem cells in birds: Opportunities and Challenges for Science and Agriculture".
  3. I recommend that the authors make some changes and improve the writing of the manuscript: 3.1. Figure 2: Add a series of boxes with bibliographic references. In Figure 2.3, add representative references in each box for each method.
  1. I do not understand the meaning of this sentence: "It is the direct result of human activity on this planet, and so it can be called current, anthropogenic, or human extinction (lines 110-11)". Please either delete or explain this sentence.
  2. More emphasis should be placed on the biotechnological implications: Although you mention applications in biotechnology, it would be helpful to go into more detail with specific examples of how iPSCs could improve agriculture or poultry production. This could make the article more accessible and relevant to readers interested in practical applications.
Comments on the Quality of English Language

Throughout the article, there are some somewhat complex sentences that could be simplified to improve flow. 

Author Response

Comment 1: The authors present a detailed and thorough review of the role of iPSCs in avian models, including their definition, history, induction mechanisms, and potential clinical and zootechnical applications, as well as their translational value. This review contributes to the existing literature by summarizing the most relevant studies and, based on this information, opens new avenues for the use of these cells in avian models and their potential applications in humans and animals.

Response 1: thank you so much for your appreciation.

Comment 2: I suggest that the title of the manuscript be changed to the following: "Induced pluripotent stem cells in birds: Opportunities and Challenges for Science and Agriculture".

Response 2: Thank you so much for your kind suggestion i have change the title according to your suggestion.

Comment 3: I recommend that the authors make some changes and improve the writing of the manuscript: 3.1. Figure 2: Add a series of boxes with bibliographic references. In Figure 2.3, add representative references in each box for each method.

Response 3: 

I have made some changes in these Figures:

i am attaching my response file you can check below.

comment 4: I do not understand the meaning of this sentence: "It is the direct result of human activity on this planet, and so it can be called current, anthropogenic, or human extinction (lines 110-11)". Please either delete or explain this sentence.

Response 4: Thank you so much for your suggestion I have removed this sentence from my article

Comment 5: More emphasis should be placed on the biotechnological implications: Although you mention applications in biotechnology, it would be helpful to go into more detail with specific examples of how iPSCs could improve agriculture or poultry production. This could make the article more accessible and relevant to readers interested in practical applications.

Response 5: 

Thank you so much for your suggestions I have added some more information regarding this point that are following:

The ability of iPSCs to help endangered animal species and conservation, to restore extinct species, and to reduce consumers dependence on animals will be deeply valued. Induced pluripotent stem cells (iPSCs) could not be differentiated from ESCs, had the property to proliferate indefinitely, and possessed the potential to give rise to all three primary germ layers. Unlike ESCs, the generation of iPSCs does not require the acquisition of embryonic tissues or oocytes. Embryos from endangered species are usually not available in sufficient numbers; induced pluripotent stem cells generated from somatic tissue are a more pragmatic source of stem cells, and fewer moral and ethical concerns surround their use. It also offers significant advantages for the use of iPSCs from domestic animals in reducing animal mortality associated with the commercial production of animal products144. Domesticated livestock deplete a large amount of natural resources through deforestation for grazing, require extensive water use, and generate immense greenhouse gas emissions. The alternative animal products made from the iPSCs could help reduce environmental damages brought by intensive farming and have sustainable economic uses. Application of iPSCs derived from domestic animals, such as cattle and swine, to produce clean meat could reduce the environmental impact caused by conventional animal husbandry. Applications of iPSCs to obtain rare animal products without harming the animals are also considered145.

Reviewer 3 Report

Comments and Suggestions for Authors

It's nicely written! Can this model be used for the treatment of virus-related pandemics?

Author Response

Comment 1: It's nicely written! Can this model be used for the treatment of virus-related pandemics?

Response 1: 

Yes, iPSC technology has significant potential for addressing virus-related pandemics. 

The use of iPSC technology to combat pandemics caused by viruses holds great promise. Recent progress in biotechnology regarding the reprogramming of cells and generating iPSCs revolutionized methodologies used for studies into the mechanisms of human disease and testing new pharmaceutical agents. This technology can also be applied to produce patient-specific models for investigations of the host–pathogen relationship and for developing novel anti-microbial and anti-viral therapies. Some applications of iPSC technology to investigate viral infections in humans include:  the modeling of human genetic predispositions to serious viral disease that includes encephalitis and serious influenza; and genetic engineering and genome editing in iPSC-derived cells obtained directly from a patient to introduce antiviral resistance148.

The ability to reprogram human induced pluripotent stem cells from donors' somatic cells has opened new avenues for the study and understanding of the basic pathophysiology of human diseases, including a growing list of viral infections like Zika Virus (ZIKV), hepatitis C virus (HCV), and Influenza virus (H1N1). Cellular reprogramming toward iPSC generation is highly cumbersome due to high costs, requirement of a great amount of resources, and the tendency of iPSCs to revert to their original somatic genotypes over time. The restricted availability of donor cells remains a formidable obstacle, especially in the creation of new pharmacological interventions against viral and other infectious diseases149. The first in vitro study that mimicked ZIKV infection in human brain cells employed two-dimensional cultures of iPSCderived neural progenitor cells and neurons [41]. That work identified that the ZIKV mainly infected neural progenitor cells, and not the induced pluripotent stem cells or neurons, causing increased cell death and interference in the regular course of the cell cycle. Another study independently showed that ZIKV infects human neural crest cells and peripheral neurons derived from stem cells in vitro and causes increased cell death and transcriptional dysregulation [42]. These examples demonstrate the application of 2D human stem cell-derived models to the study of viral tropism and cell type-specific pathogenesis150

Cell lines derived from induced pluripotent stem cells specific to individual patients allow the personalization of drugs, improving the therapeutic effectiveness. Bietti crystalline dystrophy is a rare disorder of visual impairment, and it is expected to affect about 67,000 persons w worldwide151. In cell treatments, iPSCs have emerged as a transforming resource owing not only to their capacity for development into many cell types, but also due to an inexhaustible supply and the promise of readily available cell products. Recent advancement in iPSC-derived immune cells has produced powerful Natural killer (iNK) and iT cells that, in animal models and clinical studies, displayed significant efficacy in eliminating cancerous cells152.
